# “She Has the Japanese Style”: Parenting by Japanese Immigrant Women in Korea from the Perspective of Their Children

**DOI:** 10.3390/ijerph19031494

**Published:** 2022-01-28

**Authors:** Nayoung Heo

**Affiliations:** Asian Demographic Research Institute, School of Sociology and Political Science, Shanghai University, Shanghai 200444, China; heon131@naver.com

**Keywords:** Japanese wives, parenting, Japanese Korean children, marriage immigration

## Abstract

This study looks into the parenting beliefs and behavior of female Japanese immigrants, who have stayed in Korea for more than 20 years, and attempts to explore factors for their distinct parenting style compared to their Korean counterparts. In Korea and Japan, a mother’s parenting is seen as crucial, while paternal responsibility is considered minimal. There are few studies that look into parenting patterns of Japanese immigrant women in Korea, especially from the perspective of their children. The study is based on in-depth interviews and participant observation in 2014 with 21 young Japanese-Korean adults recruited through snowball sampling. In raising children, the priorities of mothers were not academic but instead focused on children’s autonomy, wholesome personality, religiosity, various life experiences, maintaining health, and non-intervention. The mothers’ own cultural background and strong Japanese identity, limited interaction with Korean mothers or exclusive interaction with Japanese mothers, the role of the father figure as only a breadwinner, and mothers’ strong devotion to church activities tend to influence their parenting. An exploration of the mechanisms behind the differences in their styles of parenting is important before judging their parenting to be improper or maladjusted. Understanding parenting styles of families from diverse backgrounds helps to understand what society will look like in the near future.

## 1. Introduction

About 70 percent of international marriages in South Korea (hereafter called Korea) are between Korean-born males and foreign-born females, many of whom come from countries and regions including Vietnam, China, particularly the north-eastern region of China where most Korean–Chinese people (Joseonjok) reside, Japan, and the Philippines [1]. Among the major sending origins, Japan is by far the most developed country and is the origin of the fourth largest population of female marriage migrants. Despite being the oldest group of marriage migrants (and possibly the first generation of marriage migrants who came to Korea as a group), Japanese migrants, most of whom are female, have not gained substantial attention outside of being considered as a part of the female marriage migrant population as a whole. While adaptation, household issues and health aspects of marriage migrants have been topics of interest for many [2], potentially differing child rearing styles and the factors behind them have been discussed in only a few studies [3,4,5].

Among the various dimensions of child raising, past studies have focused on the education-oriented parenting prevalent throughout East Asia, especially in Korea and Japan. In these countries, maternal involvement in children’s education often includes utilizing private tutoring services (hakwon in Korea and juku in Japan) in addition to regular schooling, closely monitoring their children’s academic progress and achievements in school, and adopting an authoritarian style of parenting to regulate children’s education. While both Korea and Japan are known to have education-focused mothers, higher degrees of “educational fever” among Korean mothers are observed compared to Japanese mothers [3,6,7]. Nonetheless, few studies focus on educational involvement patterns of female marriage migrants in Korea, especially from the perspective of their children. Focusing on this population raises a number of questions. What are the parenting beliefs and behaviors of Japanese marriage immigrants after their settlement in Korea? Do they display similar patterns to those of Korean-born mothers? Parenting styles of immigrant mothers are likely to differ compared to that of Korean mothers based on their own education experience in the origin country as well as post-immigration adaptation experiences including their network formation, religious activities, and family dynamics. A few studies find that Korean mothers, compared to Japanese mothers, are more likely to show aggressive and intensive educational fever [3,6,7].

The current study attempts to explore the parenting beliefs and behavior of Japanese migrant mothers by focusing on answering the following questions. (1) Are parenting beliefs and behaviors of Japanese women different from those of Korean women? (2) What are the factors for the observed patterns? These are answered based on an analysis of interviews conducted in 2014. Twenty-one Japanese-Korean young adults, born to a Japanese-born mother and a Korean-born father, were recruited and interviewed through snowball sampling. Parenting of immigrant mothers in Korea as seen by their children is underexplored in the existing literature. Listening to the accounts of the study participants about their mothers’ parenting, the author was able to capture aspects such as their own experiences in Japan, Japanese identity, their adaptation or integration in Korea, networks with other people, influence from fathers, and religious association.

## 2. Background

In East Asia, especially Korea and Japan, the maternal role is seen as crucial, while paternal responsibility is considered minimal [8,9,10,11,12]. With no further examination of differences in parenting, Asian parenting, particularly East Asian parenting with regard to children’s education, has been seen as authoritarian, leaving little room for children’s autonomy, competition-oriented, and suggestive of “tiger-parenting” [13,14].

With reference to Baumrind’s [15] parenting typology, many scholars have characterized parenting in East Asia in an almost uniform fashion as being highly controlling [13,14]. Confucian traditions that place great value on education are deep-rooted in East Asia, especially in Korea and Japan, as demonstrated by the dependence of parents and their children on private tutoring services, in hope of higher academic achievements [16]. As there is a premium placed on educational excellence in both Korea and Japan, “educational zeal (or fever)” (The term “educational zeal (fever)” refers to extreme parental interest in children’s education commonly observed in rising industrial societies such as Korea and Japan (Seth, 2002; Sorensen, 1994)) often accelerates up to the time of the college entrance exam. Maternal involvement in Korea and Japan has been largely demonstrated by the heavy utilization of private tutoring services [3,16,17] and maternal networking, especially in Korea [18].

Nonetheless, differences and changes in behavioral patterns and in national emphases have been observed in recent years [14,19]. To be specific, being an authoritarian mother, or a “tiger mother”, may not be so typical in these two countries, as variations exist [13]. In both countries, different patterns of mothers’ involvement exist, depending on socioeconomic status [20], generation [18], and ethnicity [13]. Further, a few studies find that Korean mothers, compared to Japanese mothers, are more likely to show aggressive and intensive educational fever [3,18,21].

A marriage-immigrant mother is expected to take the primary responsibility for child rearing due to patriarchal traditions being firmly preserved [22], as in Korean native families. However, parenting of migrant women might differ in style compared to that of Korean women for multiple reasons. It can differ based on not only their own experience in the origin country, but also due to socioeconomic characteristics in the host country and the post-immigration adaptation experience [5,23]. A few studies have discussed potential behavioral differences in parenting among immigrant mothers in Korea [23,24,25]. For example, Park et al. [26] found that life satisfaction levels of mothers positively affect their parenting behavior. Life satisfaction can differ depending on how adapted they are to the new environment or on their economic circumstances. The same study also found that Korean mothers display coercive and affectionate-rational behavior more often than Japanese mothers do, implying cultural differences between the two countries.

Religion is another factor that may give rise to differences between the childrearing practices of Japanese marriage migrant mothers and Korean mothers. Compared to immigrant women from other countries such as China, the Philippines, and Vietnam, Japanese women are more likely to be passionate about their religion, as members of the Unification Church, which is the major religious institution matching Korean individuals with Japanese individuals for cross-border marriages [22]. Female Japanese marriage immigrants often assume a leading or active role in their local church and often emphasize the importance of attending the church and of maintaining religious faith to their children [27]. The time they allocate to church activities might surpass the amount of time they spend monitoring their children’s education.

While mothers are mainly in charge of household matters and raising their children, the role of fathers in Korea is often to take care of “outside” issues, namely, earning income for their family. Gradual ideational changes are occurring amongst the general public and there have been recent policy changes to support work–life balance in both Korea and Japan. However, there are still several impediments to change, including long working hours, structural barriers to work–life balance [28], and reluctance from fathers to participate in child rearing [10,11], including in many multicultural families [12,29]. The burden of childrearing on foreign-born women can be doubled if they lack a clear understanding of the cultural context or do not receive adequate support from husbands and in-laws [30].

While existing studies focus on the perspective of Japanese mothers on their parenting, the views of children on mothers’ parenting beliefs and behavior are rarely investigated. The current study attempts to describe Japanese mothers’ parenting beliefs and behavior. Further, the author aim to show the mechanisms by which the Japanese immigrant mothers’ own experiences in Japan, their networks, their participation in church activities, and the fathers’ limited involvement influence their parenting beliefs and behaviors. The current study investigates whether intensive intervention, monitoring, and emphasis on academic achievements are observed among Japanese mothers as much as among Korean mothers. Previous literature and the accounts of the study participants note the intensive controlling behaviors of Korean mothers and inconsistencies between ideal and actual behaviors of Korean mothers.

## 3. Research Questions

The current study focuses on answering the following questions.

What are the observed parenting beliefs and behaviors of Japanese marriage immigrants? Are they different from or similar to those of Korean women in terms of style and priority setting?

What are the potential factors for their beliefs and behaviors?

## 4. Data and Methods

### 4.1. Data

After IRB approval in 2014, the author recruited my interviewees through snowball sampling. An acquaintance of the author used her personal network from her local church to connect me to potential interviewees. Furthermore, the very same person introduced me to an individual who had a leading position in the Church at that time, resulting in an expansion of the pool of Japanese-Korean individuals. Moreover, the author participated in church activities to build rapport with church members, potential interviewees, and the mothers of potential interviewees to gain additional information and to become accustomed to the religious context before she began analyzing the data. In the end, the author was able to reach interviewees living in five different cities. All of the study participants are members of the Family Federation of World Peace and Unification. It is usually considered heretical by both the general public and the Protestant Church of Korea [27,31]. Nonetheless, it plays an important role in the adaptation of and connection among Japanese-Korean families in Korea.

Using the Church’s network to recruit Japanese-Korean young adults was key for the current study. It is the major religious institution facilitating cross-border marriages between Korean and Japanese people. Nearly 60% of the Japanese migrants used religious (Church) connections for the arrangement of their marriage [32]. However, the fact that all the individuals in the study sample belong to a single religion does limit the diversity of the sample.

During the period of May to July in 2014, the author interviewed twenty-one individuals in Korea, who were at least eighteen of age or older. They were born to a Japanese mother and a Korean father. All of their immigrant mothers had lived in Korea for more than 20 years. The interviews were conducted in Korean at the locations and times of their choice. All of the participants were fluent in Korean as they were born and raised in Korea. All of the interviews were audio-recorded. Each interview lasted at least for an hour or longer. A gift card worth $25 was provided to each participant as compensation for their participation. The author asked the participant at the end of each interview session to introduce her to another potential interviewee(s) if possible. This project was granted approval from the IRB board before the initial contact for recruitment was made.

It is meaningful to listen to what the children have to say about parenting behavior of their parents and how they compare their experiences with those of their peers because children are strongly affected by parenting. Children’s perceived parenting beliefs and the behavior of their mothers do not always match the mothers’ self-reported involvement. Furthermore, mothers’ self-reported behavior can distort what is actually felt by their children [33,34].

While the author does not, as of yet, directly compare the findings to the experiences of children from two-Korean-parent families, throughout the study analysis, a contextual background is provided in which these Japanese-Korean individuals are placed, using the previous literature, as well as their own perceptions of their school peers’ experiences. This is helpful to understand their comparisons of their own mothers’ parenting beliefs and behaviors to those of Korean mothers.

### 4.2. Method of Analysis

The author transcribed twenty-one audio-recorded interviews. Then, the coding process was performed with Atlas.ti. First, audio-recorded interviews were transcribed into a text file. Then, the “Initial Coding [24]” began. In this initial phase, going through each text file, the author gave each unit or part of the data a unique code. In the second phase, several quotations were bound into larger, refined codes. Next, the codes were categorized under broader categories. After multiple categorizing processes, the above-mentioned categories were then collapsed into two large themes, namely, mothers’ parenting beliefs and behavior, factors influencing mothers’ parenting beliefs and behavior, and four small themes under the factors. As explained above, the focus of the current study is to explore how the children perceived the parenting beliefs and behavior of their mothers, not the self-reported parenting behavior from the mothers. 

To check the appropriate translation of data from Korean into English, a bilingual individual was contacted, who is fluent in both English and Korean with an English education degree. A critical error in translation has not been found.

### 4.3. Demographic Characteristics of Interviewees

Prior to moving on to the analysis, a demographic profile of the interviewees provides background information so as to establish a better understanding of each interviewee (See Table 1). Names of all interviewees are pseudonyms. In terms of job status, “Military” means that the interviewee was serving at the time of interview (every Korean man has a military duty).

### 4.4. Interview Questions

Table 2 shows example questions from the interview sessions. All of the questions were generated after a careful literature review about factors that affect Korean and immigrant mothers’ parenting styles. The following list is not at all exhaustive; the list shows an outline of the questions. The author changed the form of questions for each interviewee depending on pace, interviewee’s involvement level, and quality of answers. The author sometimes added more questions when there was new information from their last answer.

## 5. Results

A few themes have been formed based on the categorizing process. The main themes are shown throughout the results section.

### 5.1. Mothers’ Parenting Beliefs and Behavior

The Japanese-born mothers of interviewees, rather than fathers, indeed played the primary role in the entire child rearing process, as expected from the existing literature. Nonetheless, their parenting style seems to differ in many ways compared to their Korean-born counterparts’. As the actual behavior of the parent is based on his or her beliefs [35], the author first demonstrates how these Japanese mothers set up priorities for their child-raising process. Furthermore, the way they are different from the “typical” Korean mothers is also explored.

Non-academic focus: To the question on priorities of their mothers in parenting, responses were surprisingly uniform in content. The most common answers were to ensure their children develop a good personality, are healthy, are committed to their religion, and have various life experiences. A 19-year-old college student, Sangjee, and many other interviewees talked about their mothers’ non-academic focuses. For example, Sangjee’s mother wanted her daughter to have good relationships, various life experiences and religious devotion. This point may be related to the tendency of Japanese mothers, despite their educational fever similar to that of Korean mothers, to not require their children to secure a lifestyle of high income or advanced education, but rather an ordinary one, especially for girls [20]. In the interviews carried out for the current study, however, this pattern was observed regardless of gender. It seems that the mothers of interviewees in general would rather their children become wholesome than especially competent. This tendency might also reflect recent ideational changes in Japanese parenting, moving towards more relaxed education [14].

There might be some inconsistencies between parental goals and actual behavior among Japanese mothers. For example, Park and Kwon [23] pointed out an inconsistency among Korean mothers: while mothers believe that securing interpersonal skills is most important in their children’s development, their actual behavior mostly emphasizes academic achievements of their children. Further, even though parents’ great emphasis on education attainment and excellence is not specific to Korea, it has been reported that “educational fever” or “educational zeal” in Korea might exceed that in Japan [3,6,21]. This is demonstrated by not only the huge investment in private tutoring but also the utilization of mothers’ networks [18]. However, the author did not find this inconsistency among the Japanese mothers.

Autonomous decision-making: One commonly observed theme in the interviews with Japanese-Korean individuals was their autonomous decision-making throughout years of schooling. Intervention by mothers in their children’s education appeared to be rare. For instance, Kiyoung recalled that his mother was supportive of whatever he wanted to do and that she did not force him to study harder. Similarly, Lia told the interviewer that even though her mother was more interested in her education than her father was, it was her mother who allowed her to develop a “self-directed learning style”. Lia made independent decisions about her education while her parents would mentally and financially support her. This is supported by the finding from Park et al. [26] that Japanese mothers tend to use less coercive and affectionate-rational behavior than Korean mothers.

Previous studies report that Korean mothers have a childrearing style that is highly focused on education. With relatively clear role divisions set up among family members, mothers are thought to be in charge of household matters, which includes their children’s education [36]. They are often referred to as authoritarian, and as constantly putting pressure on their children. This is also related to the idea that children’s success is directly related to the family’s retention of the status quo [25].

Not like “typical” Korean mothers: no pressure involved: For the mothers of the interviewees, pressuring their children was a rare occurrence. Their priorities regarding child rearing differed from those of “typical Korean mothers”, a term the interviewees frequently used in distinguishing their own mothers from the Korean-born mothers of their school peers. Furthermore, inconsistency between the ideal and reality was not observed either. Hyunsoo emphasized this point:


*Hyunsoo: My mother was not like “typical Korean mothers”. … Her priority was to make us exercise regularly and maintain good relationships. … nothing like “chee-ma-pa-ram” [a term referring to intense involvement of the mother in children’s education, literally, “the wind caused by a woman’s skirt”] … She had the “Japanese style”. Her [Japanese] friends were mostly the same. Their children do things on their own.*


The mothers’ beliefs and behavior toward their children’s education is revealed from the children’s comparison of their mothers to mothers of their school peers and from what the mothers told their children about school and studying. Lia and Nahee clearly show their mothers’ attitudes toward their education in the answers quoted below. Recall that their mothers’ priorities in childrearing do not include high-level involvement in their children’s education. Their children’s autonomy was guaranteed throughout their schooling years. In comparison, we can observe the pressure from the mothers of her school peers in Nahee’s words.


*Lia: My mother’s “educational zeal” was nothing like Korean mothers’. The education zeal of Japan is lower than that of Korea.*



*Nahee: My mom wasn’t concerned about my studies much… I even at one point wished my mom pushed me more [to study harder]. My friends were jealous of me… Their moms’ priority was to make sure they studied hard.*


Somee, another interviewee, told the interviewer that her mother “tried” sending her to private cram schools and pushing her to study. However, after some time she seemed to give up on it and let her study on her own. Mothers’ non-interventional parenting influenced my study participants’ formation of their own life principles. While they did consult their mothers for general issues related to school, relationships with friends, and religiosity, they were trusted to make major decisions about their life.

### 5.2. Potential Factors for Japanese Mothers’ Beliefs and Behavior

Cultural background: experiences in Japan: As cultural background shapes parental beliefs [37], the mothers’ negative view of the current Korean education system is based on their own experiences with the education system in their home country, Japan. A similar pattern is found in the study of Pak and Oh [38], in which, among Japanese immigrant mothers, a common criticism of the Korean education system is its extremely competitive nature. For Japanese mothers, what they observe or hear about Korean mothers’ intensive intervention in their children’s lives may seem alien or intimidating to the extreme. This point is in line with Park and Park’s [37] case study on a Japanese wife and a Korean husband. In their study, the wife expresses her frustration on the Korean education system and on the husband’s “boot camp” style of child raising. Through the accounts of interviewees, the author realized that their mothers were differentiating themselves from Korean mothers. Semi remembers her mother’s complaints about the curriculum of Semi’s school.


*Semi: My mom told me that my school curriculum doesn’t have enough physical activities classes. My mom had a swimming class and was in a baseball club in both her middle school and high school. She didn’t understand the Korean education curriculum.*


She added that her mother was more interested in the well-being of her daughter. This shows that the mothers of the interviewees were not comfortable with the intensive education-focused system of Korean schools, based on their own experience in Japan. They seemed to think that the Korean schools prevent the full development of their children, forcing them to sacrifice their childhood.

Limited Interaction with Koreans or exclusive interaction with Japanese: Among Japanese mothers, there was a common pattern of alienating themselves from Korean mothers or the mainstream. One aspect that shows this pattern is their visa status. Only two mothers have been naturalized or obtained permanent residency. This is in line with the official statistics that indicate the proportion of naturalized persons among Japanese marriage immigrants is substantially lower than those of immigrants from other countries, such as Taiwan, China or the Philippines [39]. As of 2015, only a little over 2% of Japanese marriage immigrants had obtained Korean citizenship, compared to 35% of Korean Chinese, 37% of Han Chinese, and 22% of Taiwanese marriage immigrants [40]. Having distance from the Korean mainstream might be a key factor of their differing parenting beliefs compared to those of Korean mothers. Some interviewees told the interviewer that their mothers plan to obtain citizenship when their own parents in Japan pass away. Thus, it is also possible that some mothers want to maintain ties with their family in Japan until it becomes impossible to do so.

The interviewer asked a few questions assuming that the social networks of the Japanese mothers would be based on the same ethnic group. In the Korean context, the two distinct types of mothers’ involvement in their children’s education are information-sharing with other mothers and the heavy use of private tutoring. The author asked interviewees if they have ever seen their mothers socialize with other parents and, if so, who they were and what they did together. Through these questions the author aimed to explore utilization of social networks among their mothers. The answers were strikingly similar. Below is an excerpt of Hyesun explaining her mother’s absence of “educational zeal”, unlike other mothers, noticing her minimal interest in utilizing a network.


*Hyesun: [There was] no such thing for my mom. My hometown was close to factories and there was no “educational zeal” type atmosphere in town. … [There was] no information about private academies or tutors… I was asking myself once “is it because my mom was raised in Japan that she didn’t care about hanging out with other mothers to get information?”*


Mothers of the interviewees did not really communicate with Korean mothers to exchange information regarding their children’s education. If anything, interviewees recall seeing their mothers meet with other Japanese mothers. However, those gatherings were mainly for socializing, rather than sharing education-related information. They maintained close ties with other Japanese peers in the Church or social gatherings.


*Somee: Mom would get along with Japanese mothers only. I never saw her hang out with Korean mothers. I don’t know if she was sharing information with other mothers. They talked in Japanese. But it felt more like social hangouts, not an educational-information-sharing network.*



*Sangjee: My mom didn’t communicate with Korean mothers… [She was] far from being like those Korean mothers [in terms of parenting]. She never interacted with other mothers to share information about where private academies were. I was the one to find out about hakwon [private tutoring institute].*


In contrast, Korean mothers’ networking with other mothers is a prevalent form of a mother’s involvement in their children’s education. As implied in the study of Park, Byun and Kim [17], information sharing among parents about private tutoring institutions is crucial for them to decide which private cram school they will send their children to. The quality of information shared in parental networks depends on how much time and money the parents can spend. Some mothers decide to stay home and give up on their full-time work in order to be fully involved in their children’s education [17].

While it is possible that these Japanese mothers intentionally avoided mingling with Korean mothers, there might be social barriers that do not allow them to mingle. These Japanese immigrants are likely to face challenges in adapting to a new environment, despite the fact that Korea is often considered culturally similar to Japan. They might have experienced exclusion by Koreans or voluntarily chose to associate with a Japanese community in which they feel most comfortable. Furthermore, mothers might have felt uneasy due to unfavorable portrayals of Japan in the media and public discussion of historic antagonism between Korea and Japan.

Further, the difficulty of pursuing the “Korean style” mothering, highly focused on academic excellence, is doubled when financial difficulties are added. Indeed, some mothers deliberately chose not to be naturalized for financial reasons. For instance, Somee said that her mother did not apply for citizenship even though she could have.


*Somee: My mom thinks it would be a [financial] loss if she were naturalized in Korea. If she becomes Korean, we will lose the benefit provided for multicultural families. We need to get that [from the government].*


Here, she mentioned the financial benefit that all Japanese-Korean families can receive. While the fact that they still differentiate themselves from Koreans and preserve their national identity as Japanese is one reason for not applying for citizenship, there may also be a financial factor driving reluctance to naturalize among Japanese mothers.

The study participants uniformly said that Korean-born mothers in general are extreme compared to Japanese mothers regarding their children’s education. Nevertheless, according to Kiyoung, some Japanese mothers take their children’s education very seriously and have a much stricter approach, compared to his open-minded mother. They might provide examples of mothers who are more adapted or assimilated to Korean society and have assumed the role of an education-oriented mother. An example of strict Japanese mothers in terms of education is Hyomin’s mother. Hyomin revealed that his mother was passionate about his education, often pushing him to study harder. She would obtain educational information from her circle of Japanese mothers. Moreover, his mother guided Hyomin in a very detailed way covering many dimensions of his life, such as, education, career, religiosity, and overall life principles. Below is an excerpt of Hyomin describing his mother’s parenting.


*Hyomin: Mom tailored to each of her children. She would really care about each of us in detail. She had a job and raised us carefully at the same time. She instructed me on what to do. For example, she would check on what time I went to bed, whether I studied and such. She told me I can never date a woman on my own because I have to be matched with a woman at the Church.*


Further, he thinks that his mother is Korean because his mother obtained Korean citizenship, is living in Korea, and puts Korea first before any other country.

This implies that there may be variation among Japanese mothers in terms of parenting style. In particular, obtaining citizenship might have been an important life event for Hyomin’s mother with an impact on her cultural identity. The author cannot examine this further with the current study sample as only two mothers obtained citizenship or residency. Nonetheless, it is notable that even a seemingly more assimilated Hyomin’s mother is shown to prefer relying on her own social capital to being included in a circle of Korean mothers. This could be a choice or the result of a perceived exclusion by Korean mothers. The author sensed that the mothers did not have close relationships with Korean mothers, nor did they think they could socialize with them in the first place. Further, in some sense, they were not necessarily eager to have more Korean friends.

Father figure only as a breadwinner: As expected, the involvement of fathers in the entire child rearing process seemed to be minimal. Growing up, most of the interviewees felt closer to their mothers than their fathers, and they interacted more frequently with their mothers. While talking to their mother felt natural for interviewees, they did not feel the same with their father. This tendency might be related to the tradition that the mother is responsible for taking care of the children and the father is the provider. Fathers of the interviewees were generally “too busy to nag” their children, as Sangjee explained, or did not communicate with their children frequently. Fathers in the current study, in general, were perpetuating the conventional image of a father as a breadwinner.


*Hyesun: “I am closer to my mom than my dad because we share lots of things. Also, we have similarities as women. I talk with her about every little thing. I don’t do so with my dad. He always tries to give me a solution even before I finish what I have to say”.*


Nevertheless, a few interviewees revealed that each parent was concerned with different areas of their life. For example, while their mothers mostly discussed daily concerns, things that happened in school, friends, religiosity, and church activities, their fathers were more interested in giving advice concerning career or college options. For instance, Changmo said his father was more interested in discussing major educational or life decisions while his mother talked with him about religion and daily matters.

Although there were three fathers found to be passionate about their children’s studies and career, they rarely sought educational information from parental networks or attempted to teach their children how to study effectively. Rather, they verbally pushed the children to study harder. Their interest in their children’s education, displayed by encouraging the children to obtain higher academic achievements, was particularly evident when the national college entrance exam approached. In the same way as other Korean parents, their main concern was to send their children to a prestigious college, despite the lack of discussions they had with their children. It appears that the fathers might not have realized the lack of interaction they had with their children. As most child rearing responsibilities are assumed by mothers, fathers often do not realize that they do have capabilities to provide their children with emotional warmth and attachment [40].

This detached attitude of the fathers and the relatively complete division of mothering and fathering might be important factors for the Japanese immigrant mothers in maintaining their parenting style, even after living in Korea for more than a couple of decades. A lack of social capital resources and minimal support in parenting from husbands are likely to contribute to exclusion of immigrant mothers from the competitive atmosphere of circles formed by mainstream Korean mothers.

Mothers’ devotion to church activities: All of the interviewees were official members of the Unification Church and all of the interviewees said that their mothers were highly involved in church activities. They were eager to share their worldview based on religious teachings and reminded their children to attend church regularly. For the most part, the mothers were more devoted to the Unification Church than the fathers were. This might be related to the reported tendency found in past studies of older Korean males, who have a hard time finding a Korean spouse, to use matchmaking agencies and religious institutions (particularly the Unification Church) largely for marriage purposes. Their religious beliefs may not have been as strong or prominent as those of their spouses [28]. Below is Hyesun supporting this point.


*Hyesun: Fathers in general don’t like the church because they need to spare their money for regular church offerings. The church has a different meaning to each”.*


Interviewees reported that their mothers spent more time than their fathers in the local church carrying out church activities. Some interviewees stated that their mothers were too busy with church work to be “tiger mothers”. Indeed, Japanese women and other foreign women, such as Filipinas, were often active participants in religious events and cultural performances, designers of programs, volunteers for preparing meals on Sundays and teachers giving basic religious lessons to new members of the local churches the author visited. Some mothers placed a higher value on their religion than anything else. Lia termed her learning style as “self-directed”. Sangjee and Jeemin expand on this point below.


*Sangjee: My mom was like, “you live your own life and I live my own life”. My mom would let me become whatever I want to be. If anything, my parents were anxious for me to remain religious rather than to study harder.*



*Jeemin: I had to do things myself. My parents didn’t really force me to study or go to hakwon but told me to get information myself if I needed it. My parents were always busy taking care of church-related stuff.*


Both of Jeemin’s parents have a Master’s degree and hold leading positions in the local church in one of the metropolitan cities in Korea. Her mother has also been working as a Japanese language teacher in a college, further reducing the time spent with her daughter. Jeemin emphasized that her mother respected her autonomy more than her father did. However, there was not any systematic involvement from her father either; she described her father’s verbal concerns simply as “nagging”.

Overall, the mothers’ participation in the church led their children to develop autonomous decision-making habits. Mothers participated in church activities intensively and guided their children to maintain their faith. Moreover, to their children they were living examples of following their faith and doing what is important in life. As a result, interviewees were trained to independently make decisions concerning their future, how to resolve issues, and how to seek information about schooling, college and career options. Even though one interviewee revealed that she once wished her mother were more like a “typical Korean mother” when she was little, most of the interviewees seemed to be content with the way their mothers had raised them.

## 6. Discussion and Implication

This study attempted to explore the parenting beliefs and behavior of Japanese immigrant women and the factors affecting their parenting patterns. While both Korea and Japan are known for “educational zeal” and the extensive involvement of mothers in their children’s education, the current study found that parenting priorities for the Japanese marriage immigrant mothers were making sure that their children are well-mannered and healthy, have various life experiences and religious faith, and make independent decisions. Overall, Japanese marriage migrants emphasized the autonomy of their children in making educational decisions. Academic achievements were rarely a priority. Their own experiences of growing up in Japan, limited interaction with Korean mothers and being associated with an almost exclusive group of Japanese coethnics, husbands’ limited participation in parenting, and their devotion to church activities were found to be the main factors for their distinct beliefs and behavior.

The exploration of the mechanisms behind the differences in their styles of parenting is important because without such exploration, some might judge their parenting to be ignorant, improper, or maladjusted. The current study explored common parenting beliefs and potential factors among Japanese immigrant women residing in Korea. An attempt was made to understand their distinct values in parenting by listening to children’s voices. While one can easily think that parenting patterns should be studied using a parents’ perspective, this study contributes to the literature by providing the perspective of the children.

There are limitations in the current study. Due to the difficulty of accessing the target population, the sample used in this study consists of individuals from the same religion. Future studies can explore parenting styles of various Japanese marriage migrants, such as those who are not members of the Unification Church, those who belong to other religious institutions and those who are not religious at all, and those who belong to diverse socioeconomic backgrounds. In this study, even though class information was collected, it was difficult to explore the influence of economic class. Class differentials did not appear to influence the parenting beliefs and behavior of the Japanese mothers. However, this might be because the author asked interviewees for their own perception of their economic class, not their actual household income. However, because the author interviewed the children of immigrants, not the immigrant themselves, the collection of accurate information on household income was unfeasible.

Some might argue that a mother’s educational level might be an important factor. While there were differences in educational level among the mothers, my analysis tends to find commonalities among the mothers regardless of their educational differences. However, future studies can enlarge the size of the sample and analyze the impact of educational differentials on parenting. Previous literature has found educational differences among marriage immigrant mothers from various national backgrounds, including Vietnam, Cambodia, and Thailand, and this literature has explored the impact of these differences on parenting [4]. An additional reason this study did not find educational differences in parenting beliefs and behavior may be due to the use of snowball sampling through the same church network. Immigrants who are closely associated with each other may be more likely to share similar ideas and values. Indeed, some Japanese mothers in the local church the author visited gathered on a daily basis.

There may be memory issues with my study participants. This study is retrospective in nature. The author asked participants about what they thought, heard, and felt and to share their experiences. This risk is certainly not negligible; there can be some distortion in their reorganized memories. The author understood this risk and was careful in doing a thematic analysis. Repeated validating processes were performed for the common themes in the analysis.

The current study provided insight for understanding the mechanisms behind the parenting beliefs and behavior among the Japanese immigrant mothers of 21 Japanese-Korean young adults. Japanese immigrant women and their families might obtain less attention than the other multicultural families with Vietnamese or Filipino backgrounds mainly because of their physical similarities to pass unnoticed. However, there is distinctiveness in their parenting, as this study illuminated. Moreover, while it was implied from the current study that assimilation levels of Japanese mothers can be an important factor for potential variation in their parenting, it is notable that after more than 20 years of living in Korea most of the mothers still maintain their common style of parenting. This opens up a possibility in the near future that parenting styles and maternal roles in Korea might diversify as ethnic backgrounds of second- or third-generation children become more diverse than ever. One needs to ask whether Korea has been working toward embracing parents from diverse backgrounds and understanding their cultural legacy’s impact on their children. If not too optimistic, can diversity among families and parenting styles be considered as “Korean” in the near future? Or would this take a long time with the growing presence of a conservative movement against immigrants?

## Figures and Tables

**Table 1 ijerph-19-01494-t001:** Demographic Characteristics of Interviewees.

	Joonha	Minjae	Dongmin	Hyomin	Jia	Boyoung	Jeemin	Hyesun	Nahee	Mina	Gajin
Age	19	22	20	23	20	20	21	18	22	20	24
Gender	Male	Male	Male	Male	Female	Female	Female	Female	Female	Female	Female
Birth place	Korea	Korea	Japan	Korea	Japan	Japan	Korea	Japan	Korea	Korea	Korea
Job status	Military	Military	Military	College inactive	College	College	College	High school	College	Un-employed	Em-ployed
Socioeconomic status	Middle	Middle	Middle	Middle	Middle	Lower middle	Middle	Middle	Lower	Middle	Lower
Mother’s age	48	52	51	-	52	-	53	51	54	52	61
Father’s age	54	54	52	-	53	-	52	52	54	48	63
Mother’s education	College	High school	2-year College	High school	College	High school	Master’s	College	Master’s	College	Middle school
Father’s education	College	College	2-year College	High	High	College	Master’s	High School	College	High School	High School
	**Changmo**	**Arin**	**Hyunsoo**	**Somee**	**Lia**	**Sangjee**	**Semi**	**Taeyoung**	**Kiyoung**	**Juno**	
Age	21	19	23	19	19	19	19	18	22	20	
Gender	Male	Female	Male	Female	Female	Female	Female	Female	Male	Male	
Birth place	Korea	Russia	Korea	Korea	Korea	Korea	Korea	Korea	Japan	Korea	
Job status	College	College	College inactive	College	College	College	College	College	College	College	
Socioeconomic status	Lower middle	Lower middle	Middle	Lower	Lower middle	Middle	Middle	Middle	Upper middle	Middle	
Mother’s age	52	59	60	47	52	50	52	49	51	55	
Father’s age	48	61	60	50	55	49	51	49	51	54	
Mother’s education	College	High school	College	High school	High school	High school	High school	College	College	College	
Father’s education	High School	Master’s	High School	High School	High School	2-year College	College	College	College	High School	

Note: “Socioeconomic status” indicates each interviewee’s perceived socioeconomic status of his or her family. Mother’s education: All of the mothers finished their education in Japan, except for the two who obtained their Master’s degree in Korea.

**Table 2 ijerph-19-01494-t002:** Type and examples of interview questions.

Themes	Items	Example Questions
Mothers’ prioritizing	Relationship with mother; priorities in childrearing	“What is your relationship like with your mother in general?” “What was most important to your mother when raising you?” “Could you order her priorities?”
Perceived maternal attitudes and behavior	Private tutoring experiences; decision-making as for the use of private tutoring; mothers’ networking with other mothers	“Have you ever attended a private tutoring institute for supplementary classes?” “Did your parents have any parental network for information-sharing?” “How do you compare your mother’s involvement in your education to the mothers of your peers?”
Father’s role in children’s education	Relationship with father; father’s role	“What is your relationship like with your father in general?” “What was most important to your father when raising you?”
Impact of mothers’ religious devotion on learning style of child	Mother’s religious devotion; mother’s role in the church	“Were your parents devoted to the religion?” “How important was the church to your parents?”

## Data Availability

The data presented in this study are available on request from the corresponding author. The data are not publicly available due to the participants’ privacy.

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
