# Peer review of "“She Has the Japanese Style”: Parenting by Japanese Immigrant Women in Korea from the Perspective of Their Children"

_ijerph, 2022, doi:10.3390/ijerph19031494_

Round 1

Reviewer 1 Report

An interesting study. Still, I leaves me with a lot of questions.

The mothers’ parenting started some 20 years ago. How relevant is this now?

The data were collected in 2014, which is 7 years ago. What did take you so long?

The mothers had lived in Korea for more than 20 years. When did they marry? Why did they marry with a Korean spouse? What were her immigration motives? Have none of the mothers been divorced?

Demographics I am missing. How old were the mothers and the fathers? Did the mothers have a job, at what level? And the fathers? Did the mothers speak Korean? Did the mothers go to school in Korea, how long? Which language did the mothers speak to their spouse, and to their children? Did the children speak Japanese? Is the mothers’ level of education comparable to that of the Korean mothers? And their children’s level?

How long did the interviews take? Were they in Japanese or Korean? How about the reliability of the children’s answers?

What was the mothers’ level of integration of assimilation?

The main question is: what is the effect of the absence of educational zeal and minimal interest in utilizing a network (if they had one, of course) on their children’s education and job level? This question has not been answered.

Author Response

Response to Reviewer 1' Comments

Thank You, Reviewer 1, for carefully reading my paper and giving me invaluable comments and suggestions. Below are my answers (in blue) to your questions/comments (bold).

Q1: The mothers’ parenting started some 20 years ago. How relevant is this now?

Answer: I believe that parents' parenting styles can permanently affect their children's lives. By interviewing the children of the immigrant women on their perspective about the parents' parenting, I was able to see how 20+ years of parenting might have influenced the children's value system and personal development when they were growing into young adults.

Q2: The data were collected in 2014, which is 7 years ago. What did take you so long?

Answer: I totally admit that it's old data. This data has been collected during my Ph.D. program, and I tried to reanalyze the data from a different perspective back in 2018 after I had published a paper. However, I was caught up with other responsibilities and I finally have some more time to work on this project with care. I strongly believe that this project still adds a unique perspective to understand various parenting styles of ethnic populations in Korea.

Q3: The mothers had lived in Korea for more than 20 years. When did they marry? Why did they marry with a Korean spouse? What were her immigration motives? Have none of the mothers been divorced?

Answer: Most of them were married in the 90s when many Japanese women came to Korea to marry Korean men through the Family Federation of World Peace and Unification (a religious institution in Korea). I did not include a detailed explanation in the paper about the institution due to limited space. However, I added "See" in front of the two references at line 155 so that readers can see what the institution does to match a Korean man and a Japanese woman in the references. Japanese women's motivation to marry a Korean man was mostly out of a religious mission. My sample did not include divorced couples. 

Q4: Demographics I am missing. How old were the mothers and the fathers? Did the mothers have a job, at what level? And the fathers? Did the mothers speak Korean? Did the mothers go to school in Korea, how long? Which language did the mothers speak to their spouse, and to their children? Did the children speak Japanese? Is the mothers’ level of education comparable to that of the Korean mothers? And their children’s level?

Answer: Thank you for your suggestion. I added parents' age in the table of interviewee profiles (Table 1). Instead of the occupational statuses of parents (due to limited space), I included the subjective economic status of each interviewee in the table and added a note on that below the table. The mothers spoke to their children in Korean. Many of them were fluent in Korean "like Korean mothers" according to my interviewees. Only a few interviewees spoke Japanese because they were not forced to learn or did not want to learn Japanese as a child. In terms of mother's education, both Korea's and Japan's high education systems were largely modeled by Western systems and they can be comparable. The Japanese mothers of my study participants completed up to high school or college in japan (as in Table 1). Two of them obtained their master's degrees in Korea. I added a note on this right below the table. All of my participants were educated in Korea. 

Q5: How long did the interviews take? Were they in Japanese or Korean? How about the reliability of the children’s answers?

Answer: In lines 166 and 167 in the manuscript, I added that all of the interviews were conducted in Korean because all of the participants were native Koreans (although they are multi-ethnic), according to your comment. In line 168, I added the duration of each interview. And I do recognize the reliability issue. I discussed this from lines 533 to 538 in order to make sure that I tried my best to avoid severe errors.

Q6: What was the mothers’ level of integration of assimilation?

Answer: Your point is important. However, as far as I understand, this is too much of a big theme to discuss in one paragraph or two in the current manuscript because assimilation cannot be defined with one or two dimensions. To do so, I will need to write a whole new manuscript with a different analysis with newly added data that should be additionally collected. Nonetheless, I touched on this issue throughout my manuscript, showing that Japanese mothers showed different values and behaviors in parenting compared to Korean mothers, which might indicate their potentially lower levels of assimilation. Lines 545-548 specifically touch on this point.

Q7: The main question is: what is the effect of the absence of educational zeal and minimal interest in utilizing a network (if they had one, of course) on their children’s education and job level? This question has not been answered.

Answer: It would've been very interesting to see how it affected the children's education and occupations. However, the fact that the participants were mostly in college cannot really tell us anything about the effect of the Japanese mothers' parenting style. Further, I listed my research questions from lines 139-143. Namely, "What are the observed parenting beliefs and behaviors of Japanese marriage immigrants? Are they different from or similar to those of Korean women in terms of style and priority setting? What are the potential factors for their beliefs and behaviors?" I hope you understand that my research focus was not on the effect. Further, most of my interview participants were still young adults. They were not yet searching for jobs.

I sincerely thank you for your time and detailed comments on my manuscript. I appreciate it. I hope I properly addressed your concerns in my answers and edits.

Reviewer 2 Report

The topic is quite new and interesting focusing on the rearing practices of Japanese mothers in Korea adoptive an interview-qualitative approach. However, there are several methodological concerns that should to be solved and also the text isn’t always clear, showing not a scientific soundness.

I think it should be mention in the introduction the theoretical bases of the interviews and in the methodology it should be explained the birth of the questions, how they are conducted, the interview technique and so on.

At the starting of the introduction (page 2, lines 52-55; 59-63) there is a part written in first person that is not coherent with the other text. Correct it. Also at page 3 (lines 129-136) another person change of the writing.

The same problem in data section and throughout the text. Generally, the author should explain adopting the third person introducing also his role and his role in the project.

The process of reduced dimensions in the interview should be fully described and an inter-rater agreement between two judges might be done to give more scientific solidity to this process.

The interviews’ contents are interesting and also the extracts of the narratives. I only suggest to order better them or to make a table to explain all the found themes so that the reader could follow better also the results and the extracts.

I appreciated the discussion and the conclusions.

Author Response

Response to Reviewer 2's Comments

Thank you very much for your time and comments on my manuscript. I prepared my answers in blue and your questions/comments in bold. 

Q1: The topic is quite new and interesting focusing on the rearing practices of Japanese mothers in Korea adoptive an interview-qualitative approach. However, there are several methodological concerns that should to be solved and also the text isn’t always clear, showing not a scientific soundness.

Answer: Thank you for your interest in my paper. Yes, I will try my best to address your issue throughout the text.

Q2: I think it should be mention in the introduction the theoretical bases of the interviews and in the methodology it should be explained the birth of the questions, how they are conducted, the interview technique and so on.

Answer: I understand. I included more of my theoretical base in lines from 53 to 58 to justify the need for my research. In section 4.1, I clearly stated how each interview was conducted, especially from lines 170 to 179. In section 4.4, from lines 220 to 222, I added that the questions were generated from the literature review, as you suggested, so that readers understand where my questions are coming from.

Q3: At the starting of the introduction (page 2, lines 52-55; 59-63) there is a part written in first person that is not coherent with the other text. Correct it. Also at page 3 (lines 129-136) another person change of the writing.

Answer: I changed them in the manuscript. Thanks for reading my paper carefully.

Q4: The same problem in data section and throughout the text. Generally, the author should explain adopting the third person introducing also his role and his role in the project.

Answer: I edited them throughout the whole text.

Q5: The process of reduced dimensions in the interview should be fully described and an inter-rater agreement between two judges might be done to give more scientific solidity to this process.

Answer: Thank you for your suggestion. I understand that some researchers have multiple people code their data for accuracy and consistency. While I would appreciate the time and the effort multiple people input in the process, I am not quite sure if this is the best way to validate the coding process. If two coders agree with a code, that does not necessarily mean that the code is correct. And the measurements that are used to see the inter-rater reliability tend to depend on the sample size as well. I am afraid it is not the best approach to quantify qualitative data with a quantitative method. Further, this paper is single-authored and analyzed quite some time ago, and I would not be able to recruit another person to go through the whole interview data and codes at this moment due to the time limit. If I have a chance to look at the data from a different perspective for another paper in the near future, I will try to use measurements for inter-rater agreement. 

Q6: The interviews’ contents are interesting and also the extracts of the narratives. I only suggest to order better them or to make a table to explain all the found themes so that the reader could follow better also the results and the extracts.

Answer: Yes. I see your point. I added three subtitles in section 5.1. so that readers can quickly catch the gist of each point at a glance. I think this will help readers to follow the study findings more easily. Thank you for your suggestion. 

Q7: I appreciated the discussion and the conclusions.

Answer: Thank you very much for your appreciation. I am very glad to hear that. 

I sincerely thank you for your time and comments for my manuscript. It really helped me improve my paper. I appreciate it.